# *In vitro* activity and *In vivo* efficacy of Isoliquiritigenin against *Staphylococcus xylosus* ATCC 700404 by IGPD target

**Qianwei Qu[1,2☯‡], Jinpeng Wang[1,2☯‡], Wenqiang Cui[1,2], Yonghui Zhou[1,2], Xiaoxu Xing[1,2], Ruixiang Che[1,2], Xin Liu[1,2], Xueying Chen[1,2], God'spower Bello-Onaghise[1,2], Chunliu Dong[1,2], Zhengze Li[3], Xiubo Li[4], Yanhua Li[1,2]***

**1** College of Veterinary Medicine, Northeast Agricultural University, Harbin, Heilongjiang, P. R. China, **2** Heilongjiang Key Laboratory for Animal Disease Control and Pharmaceutical Development, Harbin, Heilongjiang, P. R. China, **3** Jiangxi University of Traditional Chinese Medicine, Nanchang, Jiangxi, P. R. China, **4** Feed Research Institute Chinese Academy of Agricultural Science, Harbin, Heilongjiang, P. R. China

☯ These authors contributed equally to this work.
‡ First authorship
* liyanhua@neau.edu.cn

**Data Availability Statement:** All relevant data are within the manuscript and its Supporting Information files.

## Abstract

*Staphylococcus xylosus* (*S. xylosus*) is a type of coagulase-negative *Staphylococcus*, which was previously considered as non-pathogenic. However, recent studies have linked it with cases of mastitis in cows. Isoliquiritigenin (ISL) is a bioactive compound with pharmacological functions including antibacterial activity. In this study, we evaluated the effect of ISL on *S. xylosus in vitro* and *in vivo*. The MIC of ISL against *S. xylosus* was 80 μg/mL. It was observed that sub-MICs of ISL (1/2MIC, 1/4MIC, 1/8MIC) significantly inhibited the formation of *S. xylosus* biofilm *in vitro*. Previous studies have observed that inhibiting imidazole glycerol phosphate dehydratase (IGPD) concomitantly inhibited biofilm formation in *S. xylosus*. So, we designed experiments to target the formation of IGPD or inhibits its activities in *S. xylosus* ATCC 700404. The results indicated that the activity of IGPD and its histidine content decreased significantly under 1/2 MIC (40 μg/mL) ISL, and the expression of IGPD gene (*hisB*) and IGPD protein was significantly down-regulated. Furthermore, Bio-layer interferometry experiments showed that ISL directly interacted with IGPD protein (with strong affinity; KD = 234 μM). In addition, molecular docking was used to predict the binding mode of ISL and IGPD. *In vivo* tests revealed that, ISL significantly reduced TNF-α and IL-6 levels, mitigated the destruction of the mammary glands and reversed the production of inflammatory cells in mice. The results of the study suggest that, ISL may inhibit *S. xylosus* growth by acting on IGPD, which can be used as a target protein to treat infections caused by *S. xylosus*.

**Funding:** This work was supported by the National Key Research and Development Program of China (2018YFD0500300), Ministry of Agriculture Pig Industry System (CARS-35) and National Nature Science Foundation of China (Grant No. 31772787). The funders had no role in study design, data collection and analysis, decision to publish, or preparation of the manuscript.

**Competing interests:** The authors have declared that no competing interests exist.

## Introduction

Bovine mastitis is globally recognized as the most common and costly disease affecting dairy herds [1]. To date, more than 50 *Staphylococcus* species and subspecies have been characterized as causal agents of staphylococcal mastitis. The genus is divided into coagulase positive *staphylococci* and coagulase negative *staphylococci* (CNS) based on their ability to coagulate plasma [2]. Remarkably, CNS infection is increasingly recognized as the leading cause of clinical and subclinical dairy cow mastitis worldwide. At the same time, CNS species tend to be more resistant to antimicrobials than *Staphylococcus aureus* (*S. aureus*), and they easily develop multi-resistance [2, 3]. *Staphylococcus xylosus* (*S. xylosus*) is a CNS that is widespread in the skin of animals and man, and generally considered as a non-pathogenic bacterium [4, 5]. However, recent studies have shown that it is the main bacterial isolate in cows with cases of mastitis caused by CNS [6, 7]. Moreover, clinical isolates of *S. xylosus* have multiple drug-resistant phenotypes, which has made the treatment of dairy cow mastitis very difficult [6]. Consequently, the need to develop new drugs with novel antibacterial mechanisms is very vital in the treatment of *Staphylococcal* mastitis in dairy cows.

In order to solve the problem of bacterial resistance and develop new antimicrobial agents, researchers are exploring new frontiers including the disruption or abrogation of bacterial metabolic pathways. The relationship between bacterial metabolism and drug resistance has been well documented [8, 9]. Recent studies have shown that histidine is the product of an important nitrogen metabolism pathway in bacteria and imidazole glycerol phosphate dehydratase (IGPD) has been described as one of the key enzymes involved in its biosynthesis including acting as the first specific synthase in L-histidine synthesis pathway [10]. IGPD only exists in bacteria and plants [11, 12]. Moreover, it is associated with bacterial growth and biofilm formation [13, 14]. On the other hand, clinical treatment failure frequently occurs with the use of antibiotics due to the development of drug resistance [15]. The presence of antibiotic residues in milk significantly reduces its quality and has undesirable effects on human health [16–18]. Thus, targeting IGPD by a better and more effective alternative therapy is an important strategy in correcting the public health problem of excessive use of antibiotics and the menace of multi-drug resistant bacteria.

It has been reported that flavonoids such as baicalin can interfere with the formation of biofilm in *S. xylosus* by interfering with IGPD [19]. Flavonoids are a large class of natural compounds and have been extensively studied over the past decade for their antibacterial activity [20]. In some cases, flavonoids have showed up to six-fold stronger antibacterial activities than standard drugs in the market [21]. Isoliquiritigenin (ISL, Fig 1) is a flavonoid compound, which complies with the Lipinski's Rule of Five and has good anti-oxidation, anti-inflammatory, anti-tumor and antimicrobial activity [22–24]. Studies have shown that ISL has the potential to treat oral bacterial infections and inhibit the formation biofilms in *S. aureus* [25, 26]. At the same time, Gaur et al. observed that ISL has the potential to reverse the resistance of methicillin-resistant *S. aureus* [23]. Nevertheless, few studies have shown its effect on the growth of *S. xylosus*, the main bacterial isolates causing cow mastitis. As a natural Chinese medicine monomer, ISL is cheap, readily available, and has low toxicity [27, 28]. This study shows the antibacterial potential of Isoliquiritigenin against *S. xylosus* by regulating the expression of IGPD, and revealed the mechanism by which ISL attenuates the virulence of *S.xylosus*. In addition, the therapeutic effect of ISL against *S.xylosus* virulence was further determined in an animal mastitis model. This study will provide a scientific basis for the treatment of *S. xylosus* mastitis and the development of new drugs using Isoliquiritigenin in the future.

Fig 1. The chemical structure of isoliquiritigenin.

## Materials and methods

### Growth conditions and reagents

*S. xylosus* ATCC 700404 (from the American Type Culture Collection) and its IGPD mutant strain (obtained from the previous experiment in our laboratory by homologous recombination) were grown in Tryptic Soy Broth (TSB, Summus Ltd, Harbin, Heilongjiang, China) at 37˚C with constant shaking [14]. For biofilm culture, the mid-exponential growth culture of *S. xylosus* ATCC 700404 (Wild-type strain and mutant strain) was diluted with TSB to an optical density of 0.1 at $OD_{595}$, and 200 μL of the diluted growth culture was added to wells of a 96-well microplate (Corning Costar® 3599 Corning, NY, USA) at 37˚C for 24 h. 98% pure ISL was purchased from Shanghai Sigma-Aldrich Trading Co., Ltd. and dissolved in DMSO. ISL was prepared to a concentration of 12,800 μg/mL and double dilutions was used to do inhibitory tests *in vitro*.

### Minimum inhibitory concentration (MIC) determination

The MIC of ISL was determined in TSB using broth dilution micro method in a 96-well polystyrene microtiter plates [29]. 20 μL of ISL was obtained from the stock solution (12,800 μg/mL) and diluted to give a final concentration range from 1,280 μg/mL to 0.625 μg/mL. These were sequentially mixed with 180 μL of bacteria-containing TSB in a 96-well polystyrene microplate. Treated *S. xylosus* were incubated at 37˚C in TSB and observed for turbidity. The MIC was considered as the lowest concentration of each ISL with no visible microbial growth after 24 h of incubation [30].

### Time-dependent killing assay

The time-kill curves test was performed based on the method provided by Wang [31], with little modifications. Briefly, the *S. xylosus* suspensions were grown to an $OD_{595}$ value of 0.5 ($1 \times 10^5$ CFU/mL) in TSB. The ISL was added to the cultures to obtain final concentrations of

1/2 MIC, MIC, 2 MIC, 4 MIC and 8 MIC, respectively. A similar experiment without ISL treatment was set up to serve as the control. Cell survival was determined by plating duplicate 10-fold serial dilutions for each sample at appropriate time points and counting the colonies after incubation for 24 h at 37˚C. The detection limit was 1.0 $\log_{10}$ CFU/mL.

### Determining the effects of ISL on biofilm formation

The procedure was done in line with an earlier procedure described by Zhou [14]. *S. xylosus* ATCC 700404 and its IGPD mutant [14] strain were cultured at the mid-exponential growth phase to an optical density of 0.1 at $OD_{595}$. Cultures and sub-MICs ISL were added to each well of a 96-well microplate and a 6-well microplate (Corning Costar® 3599 Corning, NY, USA), respectively. In addition, a negative control group (bacterial culture without ISL) was setup. After incubation for 24 h at 37˚C, crystal violet staining was performed and followed by scanning electron microscopy analysis. The biofilms were obtained from bacterial cells and prepared for analysis as described by Zhou [14].

### Quantitative RT-PCR analysis

In order to investigate the effect of ISL sub-MICs on the expression of the IGPD gene (*hisB*) in *S. xylosus*, bacterial culture (mid-log growth phase) supplemented with 1/2 MIC (40μg/mL) ISL was incubated for 24 h at 37˚C with continuous shaking. Cells without ISL were used as control. Cultures were centrifuged at 10,000×g for 5 min and stored for 24 h. Thereafter, the bacterial cultures were treated with an RNase REMOVER I (Huayueyang Ltd, Beijing, China). An E.Z.N.A. ™ Bacterial RNA isolating kit was used to determine the total RNA levels. Subsequently, cDNA synthesis was performed according to PrimeScript ™ RT reagent kit protocol (Takara Biomedical Technology Co., Ltd. Beijing, China). The specific primers (*hisB* gene: *Forward*: *TACTTCTGTATCACCATT*, *Reverse*: *ACTATCTATCTCACTTGC*. house-keeping gene (*16sRNA*): *Forward*: *CGGGCAATTTGTTTAGCA*, *Reverse*: *ATTAGGTGGAGCAGGTCA*) used for the quantitative RT-PCR (Takara Biomedical Technology Co., Ltd. Beijing, China) were purchased from Takara. The quantitative RT-PCR procedure was performed as described by Yang [32].

### Expression and purification of IGPD protein and preparation of its polyclonal antibody

Purified IGPD protein and anti-IGPD polyclonal antibody were obtained from our previous studies (S1 Fig). In a nutshell, the gene (*hisB*) encoding for IGPD from *S. xylosus* ATCC700404 was designed and cloned into pET30a expression vectors (Novagen). Plasmids encoding for IGPD were transformed into *Escherichia coli* BL21 (DE3) cells (Novagen) and protein expression was induced for 5 h with 1 mM Isopropyl β-D-1-Thiogalactopyranoside at 37˚C in Luria broth. Cells were harvested and lysed by sonication. For IGPD, insoluble material was removed by centrifugation and the cell-free extracts were purified in a process involving affinity chromatography and molecular sieve chromatography. For the preparation of polyclonal antibodies, BALB/c mice (from the Experimental Animal Center of the Second Affiliated Hospital of Harbin Medical University) were injected four times subcutaneously with 1 mg of purified IGPD in Montanide ISA 206 (SEP PIC) adjuvant each time. The injections were given repeatedly at one-week interval from the beginning to the end of the experiment. Blood samples were collected from the caudal vein before the injection, and anti-sera were prepared to test the ELISA titers. The mice were euthanized 7 days after the fourth injection, and highly specific antiserum was collected.

## Western blotting

The effect of ISL on IGPD protein of *S. xylosus* was analyzed by western blotting. Firstly, the *S. xylosus* culture media (mid-log growth phase) supplemented with 1/2 MIC ISL was incubated for 24 h at 37˚C with continuous shaking. Cells without ISL served as control. Secondly, after the above cells were centrifuged, the supernatant was discarded and the cells were washed twice with PBS. Cells were then suspended in 100 μL PBS and sonicated for 3 min repeatedly for five times. Finally, 5 × loading buffer was added to the sample and boiled for 20 min. Subsequently, western blot analysis was performed according to the standard procedure [33].

## Determination of histidine content

Wild-type and mutant strains (mid-log growth phase) were inoculated into sterile TSB medium, and ISL (1/2-MIC) was added with continuous shaking in an incubator for 24 h at 37˚C. *S. xylosus* (wild-type and mutant strains) without ISL served as the control. After the above cells were centrifuged, the supernatant was discarded and the pellets were washed twice with PBS. The pellets were then suspended in sterile double-distilled water and the bacterial culture was sonicated to release histidine. The above liquid was filtered using a 0.45 μm filter, and the histidine content in the bacteria was determined by high performance liquid chromatography (HPLC) on a Waters Alliance HPLC system (Waters e2695, United States). A standard solution of 15 mg histidine (99% pure histidine—Beijing Solarbio Technology Co., Ltd.) was prepared by dissolving it in 250 mL (0.1 mol/L) hydrochloric acid. The determination of the histidine content was performed on a 5 μm Diamosil C18 column (4.6 mm × 150 mm, Japan). The chromatographic separation was carried out on a 5 μm Diamosil C18 column (4.6 mm × 250 mm, Japan) with a gradient solvent A (10 mmol/L diammonium hydrogen phosphate buffer (containing 10 mmol/L sodium 1-octanesulfonate and obtaining pH 2.0 by phosphoric acid) and solvent B (acetonitrile) as mobile phase at a flow rate of 1 mL/min. The gradient conditions were 0 to 5 min, 95% solvent A; 5 to 6 min, 95% to 86% solvent A; 6 to 15 min, 86% solvent A; 15 to 16 min, 86% to 87% solvent A; 16 to 25 min, 87% solvent A. The detection wavelength was set at 205 nm, and the injection volume was 100 μL. The column temperature was set at 8˚C. Quantification of histidine in the sonicated-treated bacterial culture was done using HPLC at 205 nm against concentration using the external standard method.

## Bio-layer interferometry (BLI)

BLI experiments were performed using an Octet system (Forte Bio) placed in PBS pH 7.4, 0.05% (v/v) Tween-20 and 1 mg/mL BSA running buffer at room temperature (25˚C). The freshly prepared IGPD protein (23 μg/mL) was coupled to the tip of a Forte Bio Octet NTA instrument. A dilution series of ISL (125,000 nM to 3,906 nM) was used to measure a dose-response curve of association and dissociation. The dissociation period was set at 5 min.

## Enzyme activity assays

*S. xylosus* cultures (mid-log growth phase) were supplemented with 1/2-MIC of ISL and incubated at 37˚C for 24 h. Cells without ISL served as control. The bacterial culture was centrifuged at 11,000 × g for 5 min. The supernatant was collected and centrifuged at 600 × g for 5 min, then the cells were sonicated on ice for 15 min. The activity of the enzyme was determined in line with a previously described stopped-assay protocol [34] with minor modifications. The reaction mixture consisted of PBS buffer with a pH of 7.4 and supernatant. The reactions were carried out at 37˚C using IGP (Santa Cruz Biotechnology, USA). The reaction

was stopped by adding sodium hydroxide after an interval of 30 s. The reaction mixture was then incubated at 37°C for 20 min to convert the product imidazoleacetol-phosphate (IAP) into an enolized form, the absorbance was read at 280 nm against a blank in a Shimadzu UV spectrophotometer. The extinction coefficient of IAP formed under these conditions is as reported previously [34].

## Molecular docking analysis of interactions between ISL and IGPD

Molecular docking method was used to verify the binding of ISL and IGPD. Homology modeling of IGPD had been done in an earlier study by Chen (*S. aureus*; PDB entry 2AE8.F) [19]. Furthermore, we also optimized the tertiary structure of IGPD as described by Ahangar [12]. The active region of the IGPD was set as a sphere (*X*:14.88, *Y*:116.26, *Z*:97.66, radius:15Å) by Discovery Studio 3.0 (DS 3.0). All the docking studies were performed by CDOCKER, which adopts the CHARMM forcefield in the DS 3.0 to dock flexible ligands into protein binding sites.

## Animal experiments

Twenty-five (25) Pregnant BALB/c mice were obtained from the experimental animal center of the Second Affiliated Hospital of Harbin Medical University. The protocol for this experiment was approved by the ethics committee of Harbin Veterinary Research Institute at the Chinese Academy of Agricultural Sciences (SYXK (Hei) 2012–2067). The care and handling of the animals were in accordance with the ARRIVE (animals in research: reporting in vivo experiments) guidelines and CONSORT (Consolidated Standards of Reporting Trials) statement [35, 36]. After 3 days of feeding without specific pathogen, they were randomly divided into 5 groups, A, B, C, D and E groups (*n* = 5).

For the establishment of a mastitis model of infection, lactating BALB/c pregnant mice, 10–12 weeks of age and weighing 30–32 g, were anesthetized using sodium pentobarbital (50 mg/kg bodyweight) and infected by injecting the canal glands with 100 μL ($1 \times 10^9$ CFU/mL) of *S. xylosus* (wild-type strain or mutant strain) suspension at the 4th breast on both sides of the abdomen counting from the head (L4 and R4). After infection for 24 h, each breast (L4 and R4) was injected with 100 μL (270 μg/kg) ISL. Animals in the control group received only 10% DMSO. All animals were monitored for 48 h to observe any pathological changes. At the end of this period (48 h), they were all euthanized by cervical dislocation. To evaluate the pathology of mastitis, mammary gland tissues isolated from the euthanized mice were fixed in 4% paraformaldehyde, stained with hematoxylin and eosin, and visualized by light microscopy. Furthermore, additional mammary gland tissues were homogenized in normal saline, after which they were partially diluted and inoculated onto TSB agar plates for the assessment of their bacterial load. The levels of the inflammatory cytokines IL-6 and TNF-α in the supernatant were measured using ELISA kits (Quanzhou Konuodi Biotechnology Co., Ltd.). Finally, total RNA was extracted from the mammary gland using RNAiso Plus and chloroform. The transcription levels of IL-6 and TNF-α were determined by real-time quantitative reverse-transcription PCR (RT-PCR). Primers for IL-6 and TNF-α and the GAPDH used as the housekeeping gene as well as conditions for RT-PCR have been described elsewhere [37, 38].

## Statistical analysis

The experiments were all performed in triplicate. Data were subjected to statistical analysis and the Wilcoxon test was conducted by SPSS17.0 software. The values were reported as mean ± standard deviation (SD). Asterisks (*) indicated statistically significant different means compared with the control group (*p < 0.05, **p < 0.01, and ***p < 0.001).

## Results

### Antimicrobial activity of ISL

The broth dilution method was used to determine the MIC, which quantified the antibacterial effect of ISL on *S. xylosus*. The MIC was 80 μg/mL.

Time-kill curves are shown in Fig 2. Compared with the control group, 1/2 MIC group did not interfere with the growth of *S. xylosus*. After 8 hours of ISL treatment at MIC level, the number of living cells decreased by 38.12%. At the same time, *S. xylosus* was completely killed at 2 MIC ISL. At the beginning (0 h), ISL had a lower CFU than the control group (except for the 1/2 MIC group), since the pretreatment took approximately 5 to 10 minutes to be completed. Moreover, the bacterial colony forming unit and the ISL concentration were dose-dependent. This may explain the decline in the initial viability of the bacteria and also demonstrates the rapid antimicrobial activity of the ISL. The results indicate that ISL may be a promising candidate for the treatment of *S. xylosus* infection.

### Determining the effects of ISL on biofilm formation

From the results, we observed that ISL sub-MICs significantly interfered with *S. xylosus* biofilm formation when compared with the control group after treatment. It is noteworthy that there was no significant interference with the formation of biofilm in the mutant strains by ISL sub-MICs (Fig 3).

### Inhibitory effect of 1/2 MIC ISL on IGPD expression in *S. xylosus*

The effect of 1/2 MIC ISL on *S. xylosus* IGPD was evaluated by quantitative RT-PCR and western blotting. The results showed that the expressions of IGPD synthesis gene (*hisB*) and IGPD

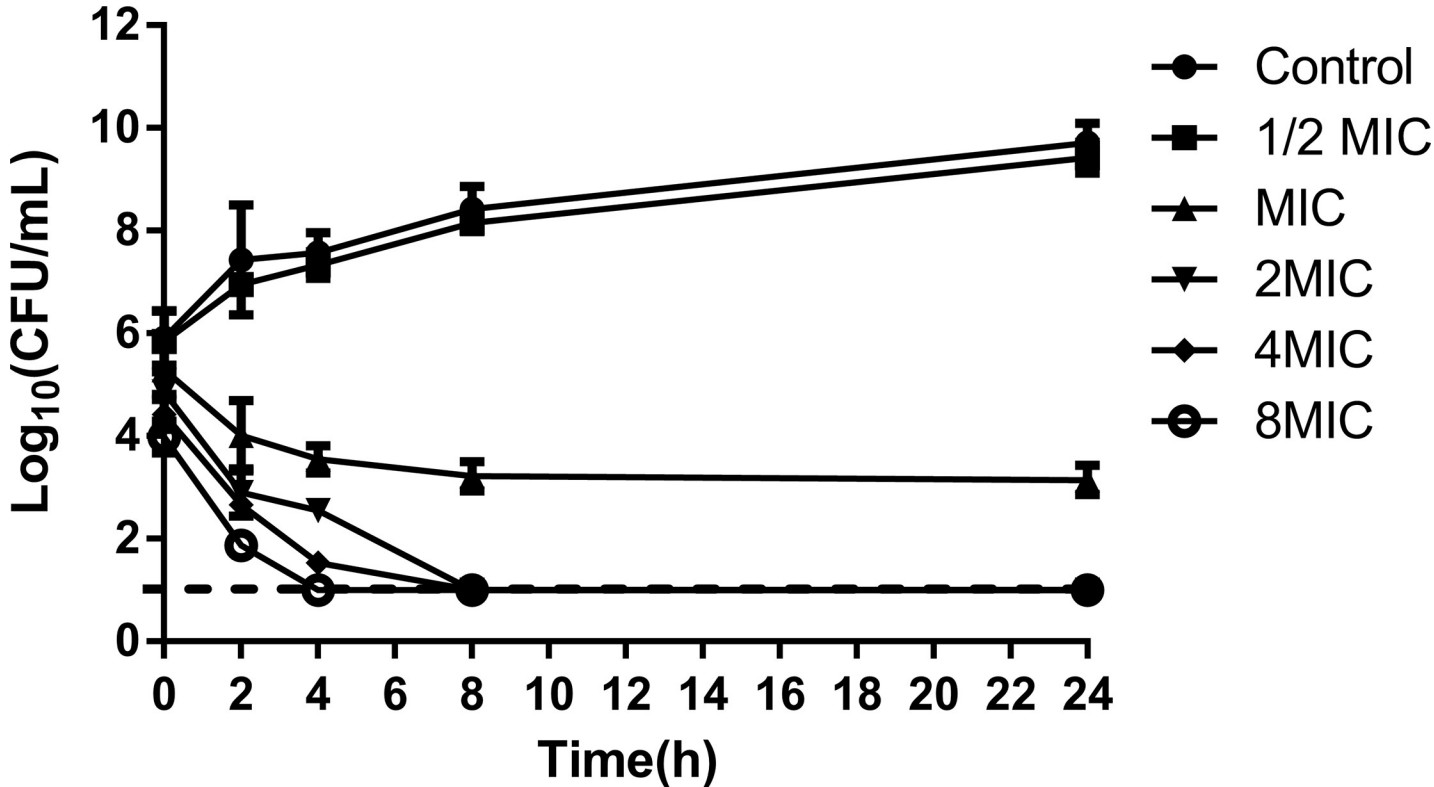

**Fig 2. Time-kill curves of ISL against *S. xylosus* cells.**

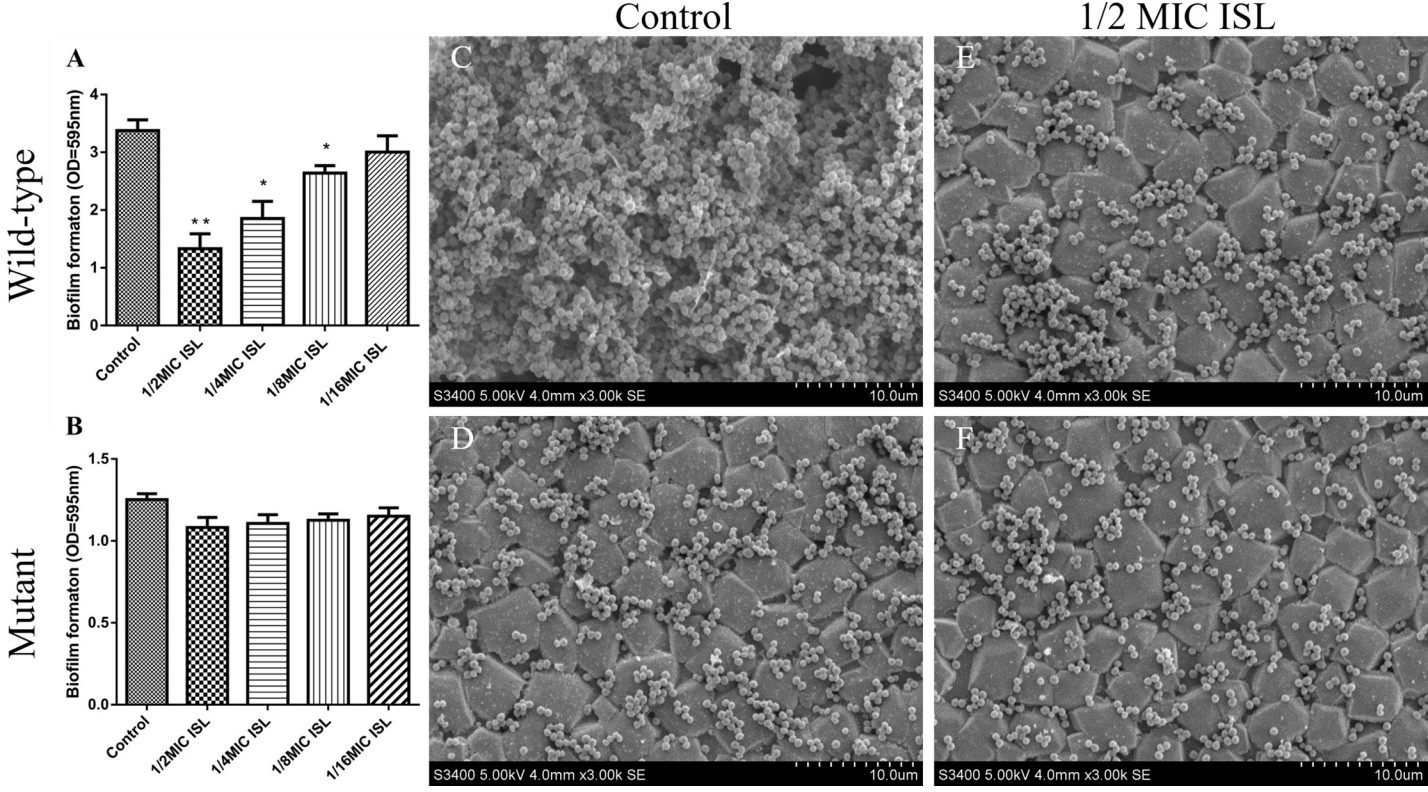

**Fig 3. Determination of biofilm formation ability (A and B) and scanning electron microscope observation (C, D, E and F) of *S. xylosus* (wild or mutant). (A)**
Effect of sub-MICs of ISL on biofilm formation by wild-type *S. xylosus* strain. Data are expressed as means ± SDs. Controls refer to the absence of ISL. Significantly different means are indicated with asterisks (*) (*$p < 0.05$, **$p < 0.01$, and ***$p < 0.001$) compared to the untreated control group. **(B)** Effect of sub-MICs of ISL on biofilm formation by mutant *S. xylosus* ATCC700404 strain. Data are expressed as means ± SDs. Controls refer to the absence of ISL. Means without asterisks (*) have no significant difference compared to the untreated control group. **(C)** Untreated wild-type strains; **(D)** Wild-type strains treated with 1/2 MIC of ISL (40μg/mL); **(E)** Untreated mutant strains; **(F)** Mutant strains treated with 1/2 MIC of ISL (20μg/mL).

protein were significantly down-regulated compared with the control group ($p < 0.05$) (Fig 4A and 4B). We speculated that ISL inhibits the *hisB*-related genes expression, thereby inhibiting the expression of IGPD proteins.

### Enzyme activity assays

This experiment tested the ability of the supernatant to bind to its substrate IGP and analyzed the activity of IGPD in *S. xylosus*. In comparison with the control group, the activity of IGPD was inhibited at 1/2 MIC ISL ($p < 0.05$) (Fig 4C).

### Determination of histidine content

HPLC chromatograms and histidine standard solution are shown in Fig 5A. The retention time of histidine in the bacterial extract was same as that of the standard (14.753 min) (Fig 5A I). Using the external standard method, the histidine content of *S. xylosus* (wild-type strain and mutant strain) with and without 1/2 MIC ISL were analyzed. It was observed that the histidine content of the wild-type strain was significantly reduced compared to the control ($p < 0.05$). Furthermore, the histidine content of the mutant strain was not significantly affected by ISL ($p > 0.05$) (Fig 5B).

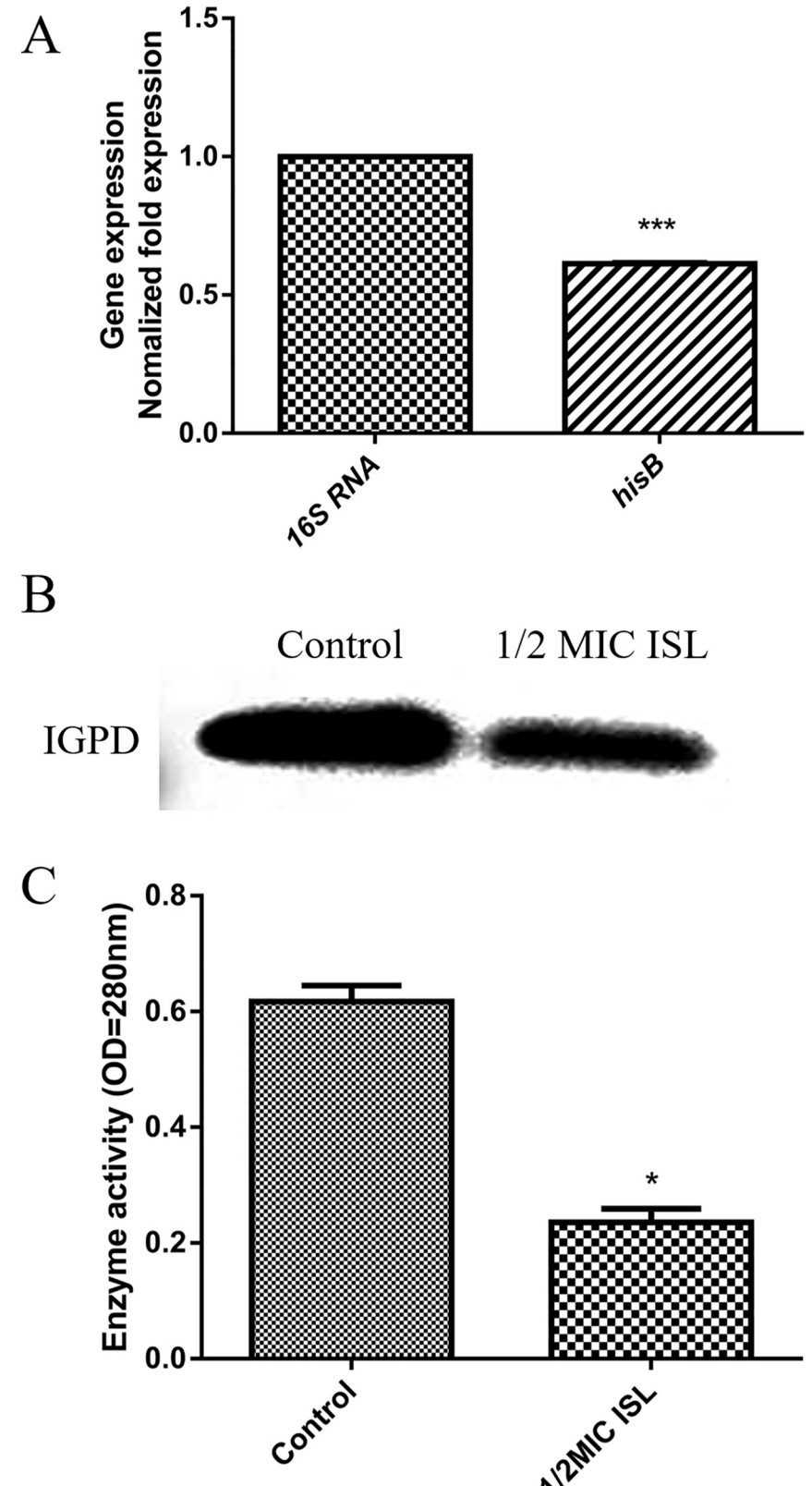

**Fig 4. Regulation of *S. xylosus* IGPD by ISL. (A)** Effect of 1/2MIC of ISL on mRNA expression of *hisB* gene in *S. xylosus*. Data are expressed as means ± SDs. Significantly different means are indicated with asterisks (*) compared to

the untreated control group (*$p < 0.05$, **$p < 0.01$, and ***$p < 0.001$). **(B)** Effect of 1/2MIC of ISL on the expression of IGPD protein in *S. xylosus*. **(C)** Determination of IGPD activity. Wild-type strain grown in the presence of 1/2-MICs of ISL. *S. xylosus* ATCC700404 served as control. Data are expressed as means ± SDs. Significantly different means are indicated with asterisks (*) compared to the untreated control group (*$p < 0.05$, **$p < 0.01$, and ***$p < 0.001$).

## BLI analysis of ISL binding to IGPD protein

We used BLI to detect the interaction between IGPD protein and ISL. The IGPD protein was first immobilized on the surface of the tip of the NTA sensor and reacted with the ISL solution. If the ISL binds to the IGPD protein, it causes a change in the surface light interference, and information on the intermolecular interaction can be obtained by analyzing the change in the light interference. Looking at Fig 6, the binding of ISL to IGPD protein reached equilibrium within 300 seconds, and it was dose-dependent as the concentration of ISL increased. The results showed that ISL interacted with IGPD protein and the dissociation equilibrium constant was given as $K_D = 234$ μM.

## Molecular docking analysis

Previous studies have shown that IGPD is a homo 24mer with two $Mn^{2+}$ and binds at each of the catalytic centers [12]. Therefore, the tertiary structure model of the constructed *S. xylosus* IGPD should contain three identical subunits containing two $Mn^{2+}$ (Fig 7A). In order to further understand the binding mode of IGPD and ISL, the molecular docking was conducted by CDOCKER. It is clear that the ISL is located at the catalytic center of the IGPD (Fig 7B). In the process of simulation, it was observed that the ISL bound to IGPD via hydrophobic, hydrogen bonds and electrostatic interactions with the amino acid residues ILE70, MET96, ASN142, and GLU162, respectively. In addition, ISL formed stronger metallic bonds with $Mn^{2+}$ (Fig 7C).

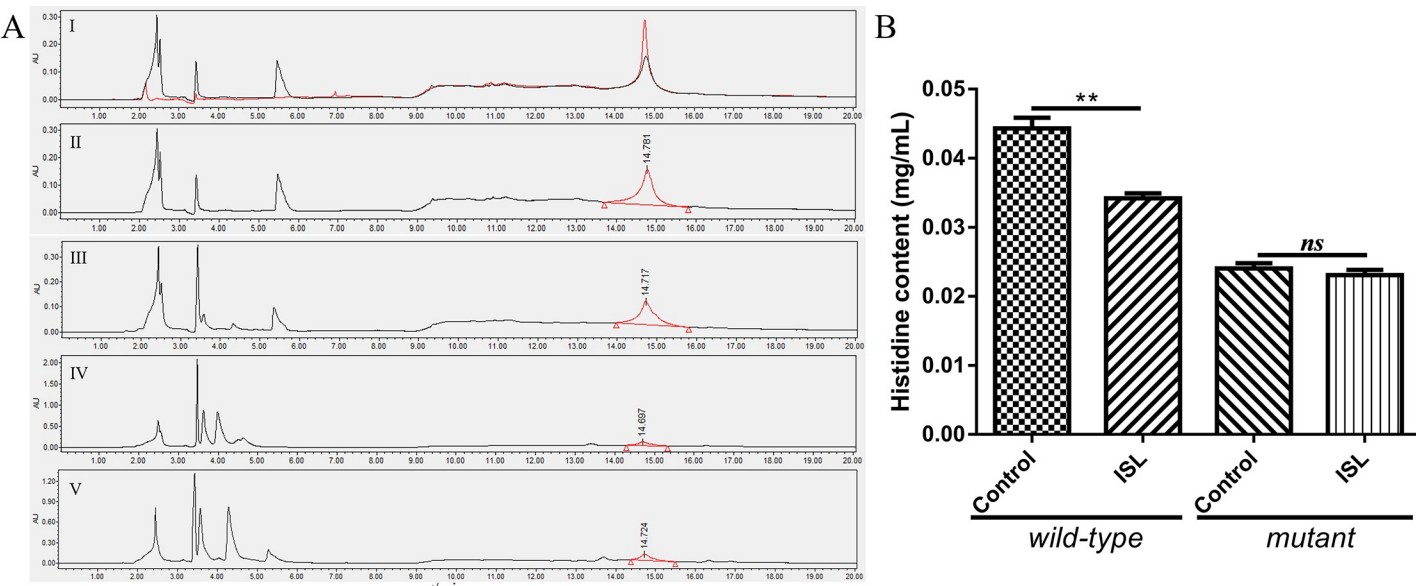

**Fig 5. Determination of *S. xylosus* histidine content. (A)** The HPLC chromatograms of bacterial extract and histidine standard solution. **I)** The HPLC chromatograms of bacterial extract (black, retention time 14.781 min) and histidine standard solution (red, retention time 14.753 min). **III and V)** *S. xylosus* (wild-type strain or mutant strain) grown in the presence of 1/2 MIC of ISL. **II and IV)** Untreated wild-type strain and mutant strain served as a control. **(B)** Peak area analysis of *S. xylosus* histidine content by HPLC. *S. xylosus* (wild-type strain or mutant strain) grown in the presence of 1/2 MIC of ISL. Untreated wild-type strain and mutant strain were used as control. Data are expressed as means ± SDs. Significant difference compared to the untreated control group are represented with asterisk (*) (*$p < 0.05$, **$p < 0.01$, and ***$p < 0.001$).

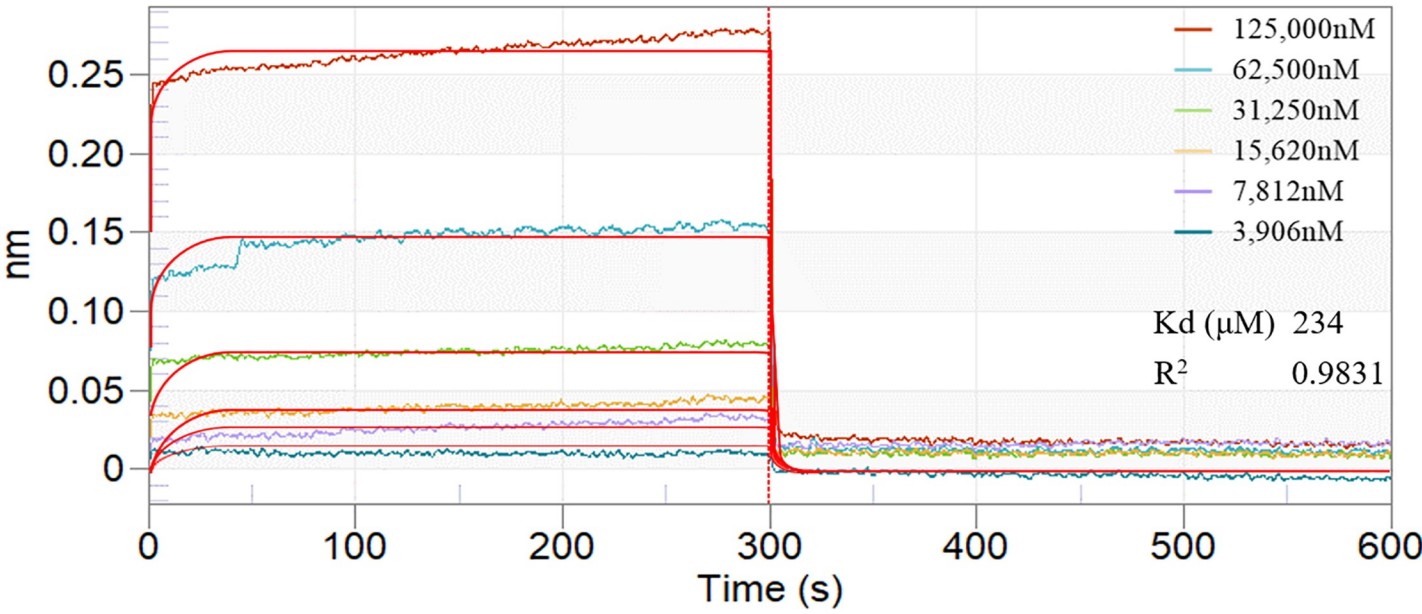

**Fig 6. Kinetic analysis by BLI of the binding of ISL to IGPD.**

## ISL inhibits *S. xylosus* in murine mastitis model of infection

In addition to the critical role of ISL against the activity of IGPD in *S. xylosus*, we also postulated that it can also inhibit the growth of *S. xylosus in vivo*. To verify this, the therapeutic effect of ISL on mastitis in mice was evaluated, and the histopathological analysis of mammary gland tissues at 48 h post-infection was performed. No inflammatory response or pathological change was detected in the negative control group (Fig 8A). While significant inflammatory responses were observed in the mice infected with *S. xylosus* (wild-type strains and mutant strains were given 10%DMSO to neutralize the effect of the solvent); responses observed included destroyed or abnormal mammary glands, dense inflammatory cells, and depletion of epithelial cells in the tissues of the mammary gland (Fig 8B and 8D). However, the ISL treatment effectively alleviated the *S. xylosus*-induced inflammatory responses and pathological changes in the infected mice, which exhibited only mild tissue injury, demonstrating a potential prophylactic effect of the ISL against *S. xylosus*-induced mastitis (Fig 8C). It is important to note that ISL did not reduce the inflammatory response in mice with mastitis induced by *S. xylosus*-mutant strain (Fig 8E). Furthermore, the bacterial counts in the mammary gland tissues indicated a significant decrease in the bacterial load in the ISL-treated mice compared with the untreated mice (wild-type strain). The bacterial load of the ISL-treated mice was not significantly different from those of the mice infected with the mutant strain (Fig 8H). In addition, the levels of mRNA and cytokines including IL-6 and TNF-α in the *S. xylosus*-infected (wild-type strains and mutant strains) mice were markedly increased compared with negative control group (10% DMSO only), whereas ISL significantly reduced the increased levels of mRNA and cytokine in the group infected with (wild-type strain) infected mice (Fig 8F and 8G and S2 Fig), there was no significant effect on the group infected with the mutant strain.

## Discussion

Cow mastitis is considered to be the most infectious disease affecting the dairy industry [39, 40]. In recent years, *S. xylosus* has become the most common CNS bacteria isolated from

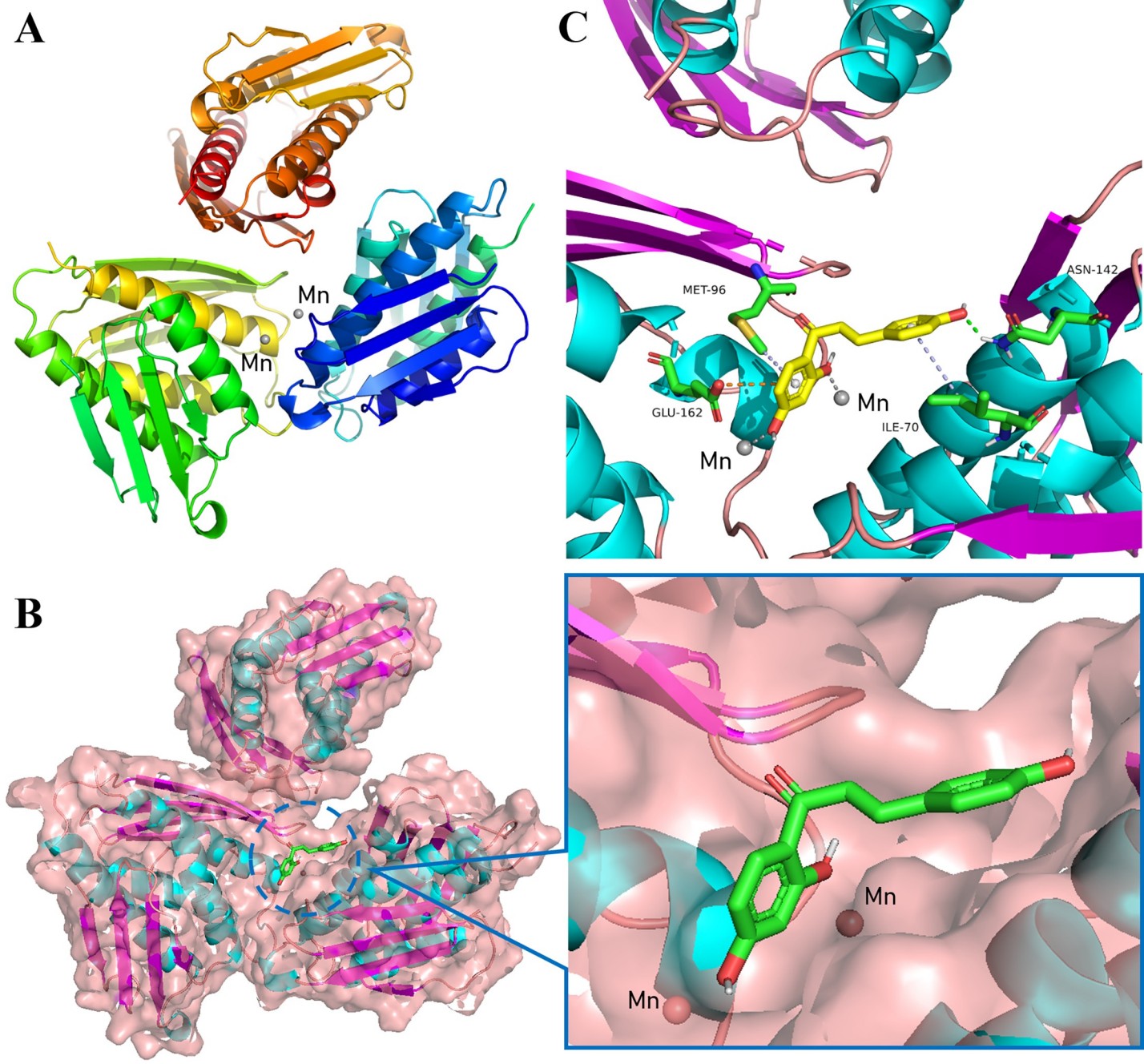

**Fig 7. Computer methodological analysis. (A)** Tertiary structure model of *S. xylosus* IGPD. **(B)** Determination of the 3D structure of the IGPD-ISL complex using a molecular modeling method. **(C)** The predicted binding mode of IGPD with the ISL is shown, with labeling of key residues in the binding sites.

clinical mastitis [2, 6, 7]. Therefore, urgent attention should be given to the treatment of cow mastitis caused by *S. xylosus*. Currently, studies have demonstrated that flavonoids have antibacterial activity and can interfere with the formation of biofilms [30, 41]. ISL is a natural compound with low molecular weight and high drug-forming property. It conforms to the Lipinski's Rule of Five and is a kind of flavanone compound with an excellent [22, 25, 42], and higher antimicrobial activity in comparison with other flavonoids [20]. Nevertheless, few

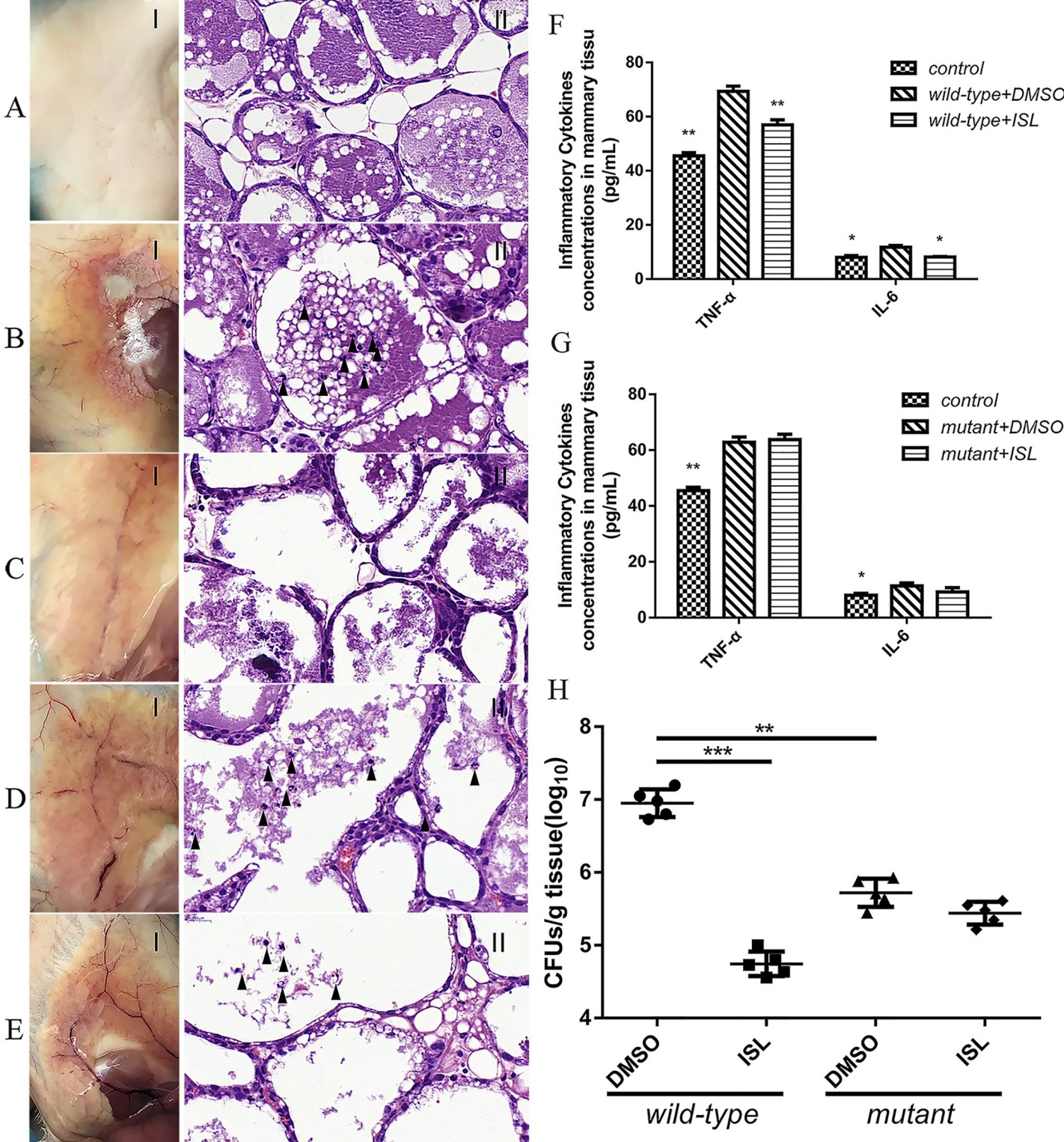

**Fig 8. ISL provides protection against *S. xylosus* mastitis in a mouse model.** Lactating BALB/c mice were infected by injecting the canal glands with $1 \times 10^9$ CFU bacterial cells. **(A-E)** The gross pathological changes (I) and histopathological analysis (II) of the mammary gland tissues at 48 h post-infection (n = 5) (Magnification: 600×). The infection group showed a large number of inflammatory factors and breast tissue destruction (arrowheads). Pathological abnormalities were significantly reduced by 250 μg/kg ISL in infected mice compared with DMSO treated mice. **(F, G)** ISL reduces the inflammatory response in infected mice. The levels of cytokines, including IL-6, and TNF-α, in the mammary gland tissues of infected mice were evaluated by ELISA (n = 5) ($^*p < 0.05$, $^{**}p < 0.01$, and $^{***}p < 0.001$). **(H)** The effect of

the ISL on the bacterial burden in infected mice. Mammary gland tissues were collected, homogenized, and plated on TSB agar plates for the assessment of the bacterial burden (n = 5) (*$p < 0.05$, **$p < 0.01$, and ***$p < 0.001$). Results are reported as mean ± SD.

studies have reported its effect on *S. xylosus*. In this study, we found that ISL had potent biological activity against *S. xylosus* (MIC was 80 μg/mL). A time-kill curve further confirmed the effect of the concentration and time-dependent killing of ISL on *S. xylosus*. In addition, TCP and SEM assays revealed that sub-MICs of ISL significantly interfered with *S. xylosus* biofilm formation *in vitro*.

Several studies have shown that IGPD is an essential enzyme in plants and microorganisms, and thus it is an attractive target for the development of herbicides and antibacterial agents, for which there are presently a limited number of biological targets [43]. Our previous studies have indicated that cefquinome and baicalin can interfere with *S. xylosus* biofilm formation by interfering with IGPD. At the same time, the interaction with IGPD will lead to a decrease in *S. xylosus* histidine content and its enzyme activity [14, 19]. Since ISL can inhibit the growth of *S. xylosus*, the sub-MIC of ISL can also interfere with its biofilm formation. So, we postulated that ISL can also affect *S. xylosus* by regulating the synthesis of IGPD or directly targeting it. First, we evaluated the relationship between ISL and IGPD gene (*hisB*) by RT-PCR. From the results, it was observed that 1/2MIC of ISL, significantly down-regulated the expression of *hisB* in the wild-type strains. We speculated that ISL may inhibit the *hisB* gene or its upstream and downstream genes, resulting in the down-regulated expression of the *hisB* gene. Subsequently, in order to verify the results, we performed western blot analysis. Similarly, 1/2 MIC ISL significantly interfered with *S.xyosus* IGPD protein expression. We speculated that ISL inhibiteds the *hisB*-related genes expression, thereby inhibiting the expression of IGPD proteins. On the other hand, histidine is the product of an important nitrogen metabolism pathway in bacteria [10], and the synthesis of histidine has been reported to indirectly affect the growth of *Corynebacterium*, *Saccharomyces cerevisiae* and *E. coli* [13, 44]. When IGPD, a key enzyme for microbial histidine biosynthesis is inhibited, its histidine content will also decrease [13, 14]. In this study, the histidine content of *S. xylosus* (wild-type strain) was significantly lowered by 1/2 MIC ISL compared with untreated group. No effect was observed in the mutant strain group treated with ISL. This may be due to the inability of the mutant strain to express IGPD, thus, not providing a target for ISL.

In order to verify whether ISL can bind to IGPD protein directly, we performed a BLI assay. BLI is an optical analysis technology which can monitor the interaction between two different molecules in real time [45]. The results showed clearly that ISL bound to IGPD protein directly with strong affinity ($K_D$ = 234 μM). Consequently, we speculated that ISL may act on the IGPD in *S. xylosus*, thus affecting the growth of the bacteria. In order to verify this conjecture, we carried out an enzyme activity experiment. The results indicated that the activity of IGPD in *S. xylosus* was significantly lower than that of the control group at 1/2 MIC of ISL. The results of BLI and enzyme activity assays revealed that ISL occupied the catalytic center of IGPD enzyme, and prevented the binding of IGPD with its substrate, then affecting *S. xylosus* (wild-type strain) enzyme activity. In order to further understand the binding mode of IGPD and ISL, we used CDOCKER for molecular docking. Previous studies have shown that IGPD is a manganese-dependent metalloenzyme composed of 24 identical subunits. Each active site consists of three subunits and two $Mn^{2+}$, the two $Mn^{2+}$ are located in the catalytic center of the active region [12, 46]. From our results, ISL bound to the amino acid residues ILE70, MET96, ASN142 and GLU162 in the IGPD active region by hydrophobic, hydrogen bonding and electrostatic interaction, respectively. In addition, ISL and the $Mn^{2+}$ forms stronger metallic bonds. It was clear that the ISL occupied the catalytic center of IGPD. Thus, results from

molecular docking further confirmed our hypothesis that ISL occupied the catalytic center of the *S. xylosus* IGPD enzyme and prevented the binding of IGPD to its substrate, which in turn affected the activity of *S. xylosus*.

Mastitis is an inflammatory disease of the breast tissue common to lactating animals and CNS has been identified as the most important group of pathogen responsible for most of the reported cases of mastitis in lactating dairy cows [2, 47–49]. Moreover, it can lead to severe breast tissue destruction and inflammation, and in most cases, treatment is not effective [6, 50]. In order to prove that ISL can effectively treat mastitis *in vivo*, we induced *S. xylosus* mastitis in mice. Histopathological examination revealed multiple inflammatory cell infiltration and mammary gland damage in the breast tissues. This was similar to what was observed in previous studies [51, 52]. In addition, cows with subclinical mastitis caused by CNS produce a local immune response in the mammary gland [5, 6], similar to a variety of inflammatory immune factors including TNF-α and IL-6 produced by Th1 and Th2-mediated immune responses in the presence of exogenous infection [53]. TNF-α is an early cytokine that plays a key role in the cascade of other pro-inflammatory cytokines and inflammatory mediators [54]. IL-6 is a pleiotropic cytokine involved in the physiology of almost every organ in an organism system [55]. Besides, TNF-α and IL-6 are crucial markers of inflammation in breast tissues [37, 56]. Therefore, the concentration of TNF-α and IL-6 in breast tissue was used as markers to indicate the occurrence of inflammation. From the results, the TNF-α and IL-6 mRNA and protein levels in the *S. xylosus* group were significantly increased. While significant inflammatory responses were observed in mice infected with *S. xylosus*, there was no observable inflammatory response or pathological change in the negative control group. Obviously, *S. xylosus* successfully induced mastitis in mice.

After *S. xylosus* infection, the production of inflammatory factors including TNF-α and IL-6 increased rapidly. Cytokines are well known to inhibit the growth of bacteria as part of host defense mechanism. In order to inhibit the growth of *S. xylosus*, Th1 and Th2 mediated immune responses produce large amounts of TNF-α and IL-6. But their overproduction can lead to systemic inflammation, which is destructive rather than protective to the host [5, 57, 58]. Therefore, inhibiting the production of TNF-α and IL-6 will concomitantly prevent inflammation. Recent studies have shown that ISL can inhibit the initial inflammatory changes on the vascular endothelium by eliminating the adhesion of monocytes [59]. In addition, ISL can down-regulate IL-6 expression in macrophages [60]. Our results revealed that ISL significantly reduced the cytokine and mRNA levels. On the other hand, ISL treatment effectively attenuated the *S. xylosus*-induced inflammatory responses and pathological changes in the infected mice which exhibited only mild tissue injury. It is worthy to note that treatment with ISL significantly reduced the bacterial load in the mammary gland tissue of the wild-type treatment group compared with the mutant strain treatment group. This indicates that ISL has an anti-inflammatory effect by regulating the expression of TNF-α and IL-6 and reversed the damaged mammary gland and mammary epithelial cells by targeting IGPD.

In conclusion, ISL down-regulated *hisB* gene expression, and thus interfered with the expression of IGPD protein in *S. xylosus*, leading to the decrease in its histidine content. Moreover, it competed with IGP in the catalytic center of IGPD, abrogating its ability to complete the dehydration reaction leading to the production of IAP and reduced the histidine content. Therefore, ISL inhibited the growth of *S. xylosus* and interfered with *S. xylosus* biofilm formation by targeting IGPD. Furthermore, *in vivo* studies confirmed that ISL effectively attenuated the damaged in mouse mammary gland tissues in the infected mice and reduced the levels of inflammatory factors produced by *S. xylosus*. To the best of our knowledge, this is the first paper reporting ISL as a potential competitive inhibitor to bind IGPD and interfere with the

pathogenicity of *S. xylosus*. Therefore, ISL may be a candidate drug for the control of mastitis in dairy cows.

## Supporting information

**S1 Fig. Preparation of IGPD. (A)** SDS-PAGE gel analysis of IGPD expression as induced by the presence or absence IPTG. Lane 1: Protein molecular mass marker; lane 2: The lysates of BL21 (DE3) cells containing pET30a-IGPD without IPTG; Lane 3: The lysates of BL21 (DE3) cells containing pET30a-IGPD with IPTG; Lane 4: The supernatants of BL21 (DE3) cells containing pET30a-IGPD with IPTG; Lane 5: The precipitates of BL21 (DE3) cells containing pET30a-IGPD with IPTG. **(B)** SDS-PAGE gel analysis of purified IGPD. The IGPD protein was purified by a Ni Sepharose 6 Fast Flow column. Lane 1: Protein molecular mass marker; lane 2: IGPD.
(TIF)

**S2 Fig. ISL reduces the expression of cytokines in infected mice. (A, B)** The expression levels of cytokines, including IL-6, and TNF-α, in the mammary gland tissues of infected mice were evaluated by qPCR (n = 5) ($^*p < 0.05$, $^{**}p < 0.01$, and $^{***}p < 0.001$).
(TIF)

**S1 File. Original images for blots and gels.**
(PDF)

## Author Contributions

**Conceptualization:** Yanhua Li.

**Data curation:** Ruixiang Che, Xueying Chen, Chunliu Dong.

**Formal analysis:** Qianwei Qu, Yonghui Zhou.

**Funding acquisition:** Yanhua Li.

**Investigation:** Xin Liu.

**Methodology:** Yonghui Zhou, Chunliu Dong.

**Resources:** God'spower Bello-Onaghise.

**Software:** Wenqiang Cui.

**Supervision:** Xiaoxu Xing, Zhengze Li.

**Validation:** Xiaoxu Xing, Zhengze Li.

**Visualization:** Wenqiang Cui.

**Writing – original draft:** Qianwei Qu.

**Writing – review & editing:** Qianwei Qu, Jinpeng Wang, Xiubo Li, Yanhua Li.

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
