## [Decision Letter · Decision Letter 0]

2 Oct 2019

PONE-D-19-21816

In vitro activity and in vivo efficacy of Isoliquiritigenin against Staphylococcus xylosus ATCC 700404 by IGPD Target

PLOS ONE

Dear Dr. Yanhua Li,

We are please to inform you that your MS is acceptable for publication with some minor corrections as suggested by reviewers. Therefore, we invite you to submit a revised version of the manuscript that addresses the points raised during the review process.

We would appreciate receiving your revised manuscript by Nov 16 2019 11:59PM. To enhance the reproducibility of your results, we recommend that if applicable you deposit your laboratory protocols in protocols.io, where a protocol can be assigned its own identifier (DOI) such that it can be cited independently in the future. For instructions see: http://journals.plos.org/plosone/s/submission-guidelines#loc-laboratory-protocols

We look forward to receiving your revised manuscript.

Kind regards,

Vivek Sharma, PhD

Academic Editor

PLOS ONE

**Journal Requirements:**

3. To comply with PLOS ONE submissions requirements, in your Methods section, please provide additional information regarding the experiments involving animals and ensure you have included details on (1) methods of sacrifice, (2) methods of analgesia, and (3) efforts to alleviate suffering.

**Additional Editor Comments (if provided):**

Dr. Yanhua Li,

The MS PONE-D-19-21816 entitled "In vitro activity and in vivo efficacy of Isoliquiritigenin against Staphylococcus xylosus ATCC 700404 by IGPD Target" has been reviewed by three reviewers and they have suggested acceptance for same with some minor correction. Additionally, as per PLOS ONE guidelines, it is mandatory for authors to submit original blots. Therefore, MS is acceptable after these corrections.

**Comments to the Author**

1. Is the manuscript technically sound, and do the data support the conclusions?

Reviewer #1: Yes

Reviewer #2: Yes

Reviewer #3: Partly

2. Has the statistical analysis been performed appropriately and rigorously? 

Reviewer #1: Yes

Reviewer #2: Yes

Reviewer #3: I Don't Know

3. Have the authors made all data underlying the findings in their manuscript fully available?

Reviewer #1: Yes

Reviewer #2: Yes

Reviewer #3: Yes

4. Is the manuscript presented in an intelligible fashion and written in standard English?

Reviewer #1: Yes

Reviewer #2: Yes

Reviewer #3: No

5. Review Comments to the Author

Reviewer #1: Authors of the article have prepared an excellent work, giving the full information of mode of action of using isoliquiritigenin (ISL), a flavonoid compound, as an drug in mice model mastitis. The excellence of the work is greater because authors gave a model of examination for future drugs. The subject of the work is pioneer, very important in veterinary, so I recommend it to publish.

In my opinion there is lack of a special subtitle/or more detailed describtion both tested microorganisms; I feel there was not distincly pointed what served as controls during the experiments. It would be nice if authors could slightly improve the article in methodological parts, although this feeling do not make the article worse.

Reviewer #2: Line 16 and Line 43: delete (S. xylosus)

Line 37: Italics Staphylococcus

Line 54: add ‘and’ after natural compounds

Line 120: Add reference of the studies

Line 230-233: Little more explanation of results is required

Line 296: Italicise in vitro

Line 315: delete et al

Reviewer #3: 1. The authors have poorly written introduction. Likewise, in the first paragraph, the authors have given the importance of S. xylosus but in the next paragraph, they suddenly shifted to (IGPD) as the key enzymes of histidine biosynthesis and in next paragraph to flavonoids. Both these aspects are different and the authors must write in a way to easily understandable to the readers. The main aim of the study should be mentioned at the end in an elaborative manner.

2. The authors have also mentioned about the expression of IGPD in pET 30a and purification as well as preparation of its polyclonal antibody in methodology. The same should be a part of results and mentioned properly. The same results could also be a part as supplementary section.

3. The authors have reported the cytokine expression of TNF-α and IL-6. Why only two cytokines have been studied. The authors must relate the results to Th1 and Th2 type immune response based on cytokine expression. The authors must report comparative cytokine expression based on qPCR as well as ELISA as the respective results are reported based on ELISA only.

4. The entire discussion provides limited insight regarding structure-function and disease status. A more direct connection should be established by making predictions and experimental testing.

5. The authors should provide the PDB identification numbers of the structures used to generate the structural models shown and data used.

6. Throughout document (nearly every sentence) - There are numerous spelling, English grammar and syntax and typo errors. This manuscript could really benefit from English editing. For example:

7. Line 15, 34: Stapylococcus should be italics

8. Line 17: can causes

9. Line 59- main bacteria isolate

10. Line 102- meaning of ½ MIC should be mentioned

11. Line 146- Use appropriate word for “The determination was performed”

12. Line 257- it also could

13. Line 302- also can

14. Line 306-307- E. coil et al

15. Line 312- can be bind

16. Line 313- remove comma

17. Line 342-343- rewrite the sentence “There were almost no inflammatory responses or pathological changes were detected in the negative control group”

18. Line 350- remove comma

19. Line 353- has a therapeutic effective against

20. Line 22, 71, 104, 131, 297, 303 and many more - remove all sticky words

6. PLOS authors have the option to publish the peer review history of their article (what does this mean?). If published, this will include your full peer review and any attached files.

Reviewer #1: No

Reviewer #2: No

Reviewer #3: No

---

## [Author Response · Author response to Decision Letter 0]

19 Nov 2019

Reviewer #1:

Authors of the article have prepared an excellent work, giving the full information of mode of action of using isoliquiritigenin (ISL), a flavonoid compound, as an drug in mice model mastitis. The excellence of the work is greater because authors gave a model of examination for future drugs. The subject of the work is pioneer, very important in veterinary, so I recommend it to publish.

In my opinion there is lack of a special subtitle/or more detailed describtion both tested microorganisms; I feel there was not distincly pointed what served as controls during the experiments. It would be nice if authors could slightly improve the article in methodological parts, although this feeling do not make the article worse.

1.Thank you for your suggestion and careful review. I have changed the subtitle to “Isoliquiritigenin inhibits the growth of S.xylosus”.

2.Thank you for your advice, I have corrected the manuscript according to your comments. I have modified the experimental method by explaining the various control groups throughout the materials and methods section of the manuscript. These corrections have been indicated in lines 100 to 101, line 109, line 207. 

3.Thank you for your valuable suggestion. I have expanded and improved the materials and methods section of the article. The modified portions are indicated in line 128, line 189, lines 214 to 218.

Revised and edited portions are written in red. I hope you will be satisfied with my answer. Thank you very much.

Thank you for giving me the chance to revise my article. I have sent this manuscript to a professional English language editor to improve the language. Thank you very much for your cautious correction. Please we will welcome any other suggestions and comments that will help to further improve the quality of the paper. Thank you once again for all your and valuable contributions.

 

Reviewer #2:

Line 16 and Line 43: delete (S. xylosus)

Line 37: Italics Staphylococcus

Line 54: add ‘and’ after natural compounds

Line 120: Add reference of the studies

Line 230-233: Little more explanation of results is required

Line 296: Italicise in vitro

Line 315: delete et al

Thank you for your valuable suggestion. I have corrected some of the grammatical errors in this article. I am very sorry for my negligence.

1. I have deleted S. xylosus in lines 18 and 44.

2. I have italicized the Staphylococcus in line 37.

3. I have added ‘and’ after natural compounds in line 63.

4. I have placed the reference picture (Fig. S1) in the supplementary material in lines 127 to 128.

5. I have added some explanations of the results in lines 243 to 245.

6. I have Italicized ‘in vitro’ in line 308.

7. I have deleted ‘et al’ from line 325.

Revised and edited portions are written in red. I hope you will be satisfied with my answer. Thank you very much.

Thank you for the time spent in correcting my article. This has given me the opportunity to modify the work. I am grateful for all your remarks concerning the grammatical errors in the article and subsequently, the manuscript has sent to a professional English language editor for correction. I have modified the paper in line with to all your comments. Please I will be very grateful if other portions that need further improvement is brought to my notice for further correction. Thank you once again for your time.

 

Reviewer #3: 

1. The authors have poorly written introduction. Likewise, in the first paragraph, the authors have given the importance of S. xylosus but in the next paragraph, they suddenly shifted to (IGPD) as the key enzymes of histidine biosynthesis and in next paragraph to flavonoids. Both these aspects are different and the authors must write in a way to easily understandable to the readers. The main aim of the study should be mentioned at the end in an elaborative manner.

2. The authors have also mentioned about the expression of IGPD in pET 30a and purification as well as preparation of its polyclonal antibody in methodology. The same should be a part of results and mentioned properly. The same results could also be a part as supplementary section.

3. The authors have reported the cytokine expression of TNF-α and IL-6. Why only two cytokines have been studied. The authors must relate the results to Th1 and Th2 type immune response based on cytokine expression. The authors must report comparative cytokine expression based on qPCR as well as ELISA as the respective results are reported based on ELISA only.

4. The entire discussion provides limited insight regarding structure-function and disease status. A more direct connection should be established by making predictions and experimental testing.

5. The authors should provide the PDB identification numbers of the structures used to generate the structural models shown and data used.

6. Throughout document (nearly every sentence) - There are numerous spelling, English grammar and syntax and typo errors. This manuscript could really benefit from English editing. For example:

7. Line 15, 34: Stapylococcus should be italics

8. Line 17: can causes

9. Line 59- main bacteria isolate

10. Line 102- meaning of ½ MIC should be mentioned

11. Line 146- Use appropriate word for “The determination was performed”

12. Line 257- it also could

13. Line 302- also can

14. Line 306-307- E. coil et al

15. Line 312- can be bind

16. Line 313- remove comma

17. Line 342-343- rewrite the sentence “There were almost no inflammatory responses or pathological changes were detected in the negative control group”

18. Line 350- remove comma

19. Line 353- has a therapeutic effective against

20. Line 22, 71, 104, 131, 297, 303 and many more - remove all sticky words

1.Thank you for your suggestion and careful review. I have expanded and improved the introduction of this article to strengthen the connection between paragraphs. This is indicated in lines 46 to 63. I have also mentioned the main purpose of the study in a relatively detailed manner in lines 72 to 77.

2.Thank you for your advice, I have corrected the manuscript according to your suggestion. I have reported the protein expression and purification results as one of the supplemental materials (Fig. S1).

3.Thank you for such a careful review.

a. I have explained the critical roles of TNF-α and IL-6 as markers of inflammation in breast tissue in lines 355 to 357.

b. In lines 366 to 371, I have linked Th1 and Th2 type immune responses to cytokine expression and conducted in-depth analysis to buttress this claim. 

c. I have edited the qPCR test in lines 214 to 218, and the results are shown in the supplementary materials section (Fig. S2).

4.Thank you for your suggestion and careful review. In lines 339 to 344, I have improved my discussion of structure-function and disease status with a more detailed explanation, linking postulations with confirmatory tests.

5. I have added the PDB identification number to line 189.

6.Thank you for your valuable suggestion. I am very sorry for my negligence. I have corrected some of the syntax errors in this article.

7. I have Italicized ‘Staphylococcus’ in lines 17 and 37.

8., I have edited ‘can causes’ and changed it to ‘linked it with cases of’ in line 19.

9. In line 71, I have replaced ‘main bacteria isolate’ to ‘main bacterial isolates’.

10. I have replaced ‘1/2 MIC’ with ‘sub-MICs’ in line 113.

11. ‘The determination was performed’ with ‘The experiment to determine the histidine content’ in line 159.

12. I have replaced ‘it also could’ with ‘it can also ‘in line 276.

13. I have replaced ‘also can’ with ‘interfered’ in line 321.

14. I have deleted et al from line 325.

15. I have changed ‘can be bind’ to ‘can bind’ in line 330.

16. I have removed the comma from line 331.

17. I have deleted ‘There were almost no inflammatory responses or pathological changes were detected in the negative control group’ and replaced it with ‘There were no observable inflammatory responses or pathological changes in the negative control group’ in lines 363 to 365.

18. I have deleted the comma from line 376.

19. I have deleted ‘has a therapeutic effective against’ from lines 378 to 380 and replaced it with ‘This indicates that ISL has an anti-inflammatory effect by regulating the expression of TNF-α and IL-6 and reversed the damaged mammary gland and mammary epithelial cells by targeting IGPD’.

20. I have removed all sticky words including lines 24, 82, 115, 144, 316 and 322.

Revised and edited portions are written in red. I hope I have been able to correct all the issues you raised in the original manuscript satisfactorily. Thank you once again for your valuable contributions.

Thank you for giving me the opportunity to revise my article. I was particularly concerned about the final decision of this article. There may be some deficiencies in my paper, but I will be very careful to modify and improve it. I am grateful for all your remarks concerning the grammatical errors in the article. I have sent this manuscript to a professional English language editor to improve the language. I have made cautious effort to modify the paper according to all your comments. Please feel free to inform me of any portion that need further correction. Thank you once again for your time.

---

## [Decision Letter · Decision Letter 1]

25 Nov 2019

In vitro activity and in vivo efficacy of Isoliquiritigenin against Staphylococcus xylosus ATCC 700404 by IGPD Target

PONE-D-19-21816R1

Dear Dr. Yanhua Li,

We are pleased to inform you that your manuscript has been judged scientifically suitable for publication and will be formally accepted for publication once it complies with all outstanding technical requirements.

With kind regards,

Vivek Sharma, PhD

Academic Editor

PLOS ONE

Additional Editor Comments (optional):

Dear Dr. Yanhua Li,

I am pleased to inform you that MS PONE-D-19-21816R1 entitled "In vitro activity and in vivo efficacy of Isoliquiritigenin against Staphylococcus xylosus ATCC 700404 by IGPD has been accepted for publication.

Thank you for submitting your work to PLOS One. We hope you consider us again for future submissions.

Kind regards

Reviewers' comments:

Reviewer's Responses to Questions

**Comments to the Author**

1. If the authors have adequately addressed your comments raised in a previous round of review and you feel that this manuscript is now acceptable for publication, you may indicate that here to bypass the “Comments to the Author” section, enter your conflict of interest statement in the “Confidential to Editor” section, and submit your "Accept" recommendation.

Reviewer #3: All comments have been addressed

2. Is the manuscript technically sound, and do the data support the conclusions?

Reviewer #3: Yes

3. Has the statistical analysis been performed appropriately and rigorously? 

Reviewer #3: Yes

4. Have the authors made all data underlying the findings in their manuscript fully available?

Reviewer #3: Yes

5. Is the manuscript presented in an intelligible fashion and written in standard English?

Reviewer #3: Yes

6. Review Comments to the Author

Reviewer #3: The manuscript is revised and modified now. All the comments are addressed point wise and the present form of manuscript is acceptable.

7. PLOS authors have the option to publish the peer review history of their article (what does this mean?). If published, this will include your full peer review and any attached files.

Reviewer #3: No

---

## [Editor Report · Acceptance letter]

12 Dec 2019

PONE-D-19-21816R1 

 *In vitro* activity and *in vivo* efficacy of Isoliquiritigenin against *Staphylococcus xylosus* ATCC 700404 by IGPD target

Dear Dr. Li:

I am pleased to inform you that your manuscript has been deemed suitable for publication in PLOS ONE. Congratulations! Your manuscript is now with our production department. 

With kind regards,

on behalf of

Dr. Vivek Sharma 

Academic Editor

PLOS ONE